# Causality-driven Hierarchical Structure Discovery for Reinforcement Learning

**Shaohui Peng**[1,2,3]   **Xing Hu**[1]   **Rui Zhang**[1,3]   **Ke Tang**[4]
**Jiaming Guo**[1,2,3]   **Qi Yi**[1,3,6]   **Ruizhi Chen**[2,5]   **Xishan Zhang**[1,3]
**Zidong Du**[1,3]   **Ling Li**[2,5]   **Qi Guo**[1]   **Yunji Chen**[1,2,†]

[1]SKL of Processors, Institute of Computing Technology, CAS
[2]University of Chinese Academy of Sciences   [3]Cambricon Technologies
[4]Department of Computer Science and Engineering, Southern University of Science and Technology
[5]SKL of Computer Science, Institute of Software, CAS
[6]University of Science and Technology of China
{pengshaohui18z, huxing, zhangrui, guojiaming18s,
zhangxishan, duzidong, guoqi, cyj}@ict.ac.cn,
tangk3@sustech.edu.cn, yiqi@mail.ustc.edu.cn,
{ruizhi, liling}@iscas.ac.cn

## Abstract

Hierarchical reinforcement learning (HRL) effectively improve agents' exploration efficiency on tasks with sparse reward, with the guide of high-quality hierarchical structures (e.g., subgoals or options). However, how to automatically discover high-quality hierarchical structures is still a great challenge. Previous HRL methods can hardly discover the hierarchical structures in complex environments due to the low exploration efficiency by exploiting the randomness-driven exploration paradigm. To address this issue, we propose CDHRL, a causality-driven hierarchical reinforcement learning framework, leveraging a causality-driven discovery instead of a randomness-driven exploration to effectively build high-quality hierarchical structures in complicated environments. The key insight is that the causalities among environment variables are naturally fit for modeling reachable subgoals and their dependencies and can perfectly guide to build high-quality hierarchical structures. The results in two complex environments, 2D-Minecraft and Eden, show that CDHRL significantly boosts exploration efficiency with the causality-driven paradigm.

## 1 Introduction

Reinforcement learning (RL) enables the intelligence of the agents by learning to take actions in the environments for maximum rewards [32, 23, 21]. For complex tasks with large state spaces and sparse delayed rewards [14], hierarchical reinforcement learning (HRL) extends RL methods' successes by discovering beneficial hierarchical structures, which helps the agent train multiple levels of policies and explore efficiently using high-level actions [26].

The major challenge of HRL methods is how to discover high-quality hierarchical structures (e.g., subgoals, skills, or options)[1]. A straightforward method is to manually define the hierarchical structures based on task-specific prior knowledge. However, in observing that the hand-crafted

---

† Corresponding author.

[1]In this paper, the hierarchical structure refers to the subgoal hierarchy in the environment.

36th Conference on Neural Information Processing Systems (NeurIPS 2022).

hierarchical structures require human expertise and yield in low generality, some researchers propose methods to discover subgoals automatically [24, 19, 37, 25]. These methods focus on how to discover subgoals from experience but follow an inefficient randomness-driven exploration paradigm to obtain experience. Specifically, the agent explores the environment through the randomness of the policy, which is implemented by raising the entropy of the policy or adding random noise to actions, and then discovers subgoals from the collected experience. Randomness-driven exploration paradigm hardly discovers the high-quality hierarchical structure in complicated environments, such as Minecraft, because of low exploration efficiency, thus limiting the HRL method's performance.

To address the above issue, we propose a Causality-Driven Hierarchical Reinforcement Learning (CDHRL) framework which leverages the causality in the environment as the guidance to discover high-quality hierarchical structures. The key insight is that causality in a system, which explains the dependencies among objects, states, events, or processes, can be naturally used to describe a final goal with sub-goals structurally. Therefore, in CDHRL, we use the causalities among environment variables to guide the discovery of the subgoal hierarchy. Such a causality-driven paradigm avoids the inefficient random exploration process, and significantly enhances the efficiency of discovering hierarchical structures in complicated environments. Moreover, the generated high-quality hierarchical structure, which consists of subgoals and their dependency, is more instructive and efficient during training than the traditional fully-connected hierarchical structure.

In implementing CDHRL, we utilize an iterative boosting framework to progressively discover the causality and construct the hierarchical structure, since it is difficult to discover all causalities in complicated environments directly. The proposed CDHRL consists of two processes, causality discovery and subgoal hierarchy construction. Specifically, causality discovery refers discovery of causality between environment variables. In subgoal hierarchy construction, we find reachable subgoals, which are changes of controllable environment variables, and leverage causality to construct dependency of subgoals. The above two processes can iteratively boost each other: On one side, we can transform discovered causality between environment variables into the subgoal hierarchy. On the other side, we can utilize trained subgoal-based policies to enhance the efficiency of causality discovery. Taking a simplified example in Minecraft play, stonepickaxe (SP) and ironore(IO) are environment variables. After the agent learned subgoals of getting SP, CDHRL could control SP's distribution to discover causality from SP to IO, which indicates that the number of stonepickaxes affects the probability of acquiring ironore. Then, CDHRL adds subgoals about the effect variable IO to the subgoal hierarchy and trains corresponding subgoal-based policies. After that, we in turn discover further causality with IO as cause efficiently by utilizing subgoals about IO to change IO's distribution and collect intervention data. Such a causality-driven discovery paradigm makes the two processes, causality discovery and subgoal hierarchy construction, iteratively promote each other.

We verified our method in two typical complex tasks, including 2d-Minecraft [30] and simplified sandbox survival games Eden [4]. The results show that CDHRL discovers high-quality hierarchical structures and significantly enhances exploration efficiency. Compared to the state-of-the-art HRL method HAC [19] and HRL enhanced by curriculum learning [27], our method significantly outperforms existing studies in terms of both final performance and learning speed.

## 2 Related Works

### 2.1 Hierarchical Reinforcement Learning

Discovery of hierarchical structure plays an essential role in the Hierarchical Reinforcement Learning (HRL) algorithm. Some methods reduce the difficulty of discovering the hierarchical structure by adding artificial prior, such as sub-tasks dependency graph [30] and manually defined subgoals [10, 33]. Others may manually restrict the form of hierarchical structure to reduce the search space [12, 11, 6]. By introducing prior information, these methods perform well on specific tasks. However, it not only requires solid expert knowledge but also sacrifices versatility. Our work does not introduce downstream task-specific information, and thus has a broader application.

Some researchers propose methods to discover subgoals automatically. Vezhnevets et al. [34] and Nachum et al. [24] restrict the goal space as state space and train goal-based hierarchical policy end-to-end. Levy et al. [19] enable the multi-level hierarchical structure with hindsight corrections. Some methods discover hierarchical structure by optimizing additional objectives. Zhang et al. [37] assume that the subgoals should be near to the current state. Nachum et al. [25] claim that the optimal

policy based on subgoals should have minimal loss than that based on primitive actions. Pitis et al. [27] consider the training progress for finding the next desired subgoal. Existing methods focus on discovering subgoals from data that is randomly collected. They are often applied in easy-to-explore environments like Maze [2] or Robot [9], but hard work in a complicated environment due to the low exploration efficiency of their randomness-driven exploration paradigm. CDHRL follows the causality-driven exploration paradigm, which is more suitable for complicated environments.

## 2.2 Causal Reinforcement Learning

Causal reinforcement learning is a research direction that combines the causal effect with reinforcement learning. Pitis et al. [28] leverages local causal structures to improve the sample efficiency of off-policy RL. Méndez-Molina et al. [22] makes a trade-off between exploration and exploitation based given causal graph. Corcoll and Vicente [7] distinguishes the controllable effect of the agent by conducting counterfactual detection. Sontakke et al. [31] and Seitzer et al. [29] discover simple causal influences to improving the efficiency of reinforcement learning. Guo et al. [13] learns the causality between the hindsight effect variables and estimated values to decrease the variance in the policy training phase. Most of them utilize predefined causality graph as prior knowledge, or detect one-step causality to enhance RL method. However, none of them automatically discover complex causality graph in the environment to guide the exploration of hierarchical structure. Causal discovery methods [15, 16, 18] discover causality between variables through the data, which is an important issue in causal learning. Our method introduces a causal discovery-based exploration dramatically enhances the discovery efficiency of hierarchical structures in complex environments.

## 2.3 Relational Reinforcement Learning

Relational Reinforcement Learning (RRL) represents entities, actions, and policy by relational language to combine relation learning or reasoning with RL[8], which shares similar motivation with our method. Zambaldi et al. [36] adopt self-attention mechanism upon entities-centric representation to implements relation learning . Wang et al. [35] learns particular relations using a graph neural network to promote zero-shot performance on continuous control tasks. Chitnis et al. [5] and Kokel et al. [17] both learn relation model and planner in first-order language described environment. RRL aims to combine relation induction and reasoning with RL to promote policy generalization, which often works in structural environments with a clear definition of logical predicates. Our method adopts causality, which is more simple and common than logical relation, to guide the direction of subgoals discovery and thus enhance its efficiency in complicated environments.

# 3 Preliminary

## 3.1 Subgoal-based Markov Decision Process (MDP)

Hierarchical reinforcement learning that adopts subgoals as hierarchical structure extends the MDP framework by introducing the subgoal space $\mathcal{G}$. Subgoal-based MDP is a six-tuple, $(\mathcal{S}, \mathcal{G}, \mathcal{A}, T, r, \gamma)$, including a state set $\mathcal{S}$, a subgoal space $\mathcal{G}$, and an action set $\mathcal{A}$. The transition probability function $T(s, a, s')$ represents the probability of transition from $s$ to $s'$ with executing action $a$. $r(s, g, a, s')$ represents the goal-reaching reward function that indicates whether the agent achieves the subgoal $g$ in transition $(s, a, s')$. And $\gamma$ is the discount rate. The target of reinforcement learning is to find a subgoal-based policy $\pi : \mathcal{S}, \mathcal{G} \rightarrow \mathcal{A}$ that maximizes the accumulated rewards $R_\pi(s, g) = E[\sum_{n=0}^{\infty} \gamma^n R_{t+n+1} | s_t = s, g_t = g]$. Our proposed framework aims to pre-train the subgoal-based policy for downstream sparse reward tasks.

## 3.2 Causal Discovery

Causal discovery aims to infer causality through variables data under causal modeling and assumptions. There are two key parts of it: how to represent causality and how to obtain the data for discovering causality.

**Structural Causal Model (SCM)** is a kind of model that describes the system's causality. SCM over a finite number M of random variables $X_i$ includes a directed acyclic graph (DAG) $C \in \{0, 1\}^{M \times M}$ as the causality graph, and a set of generating functions $f_i$ of variables:

$$X_i := f_i(X_{pa(i,C)}, N_i), \quad \forall i \in \{0, \dots, M - 1\}. \tag{1}$$

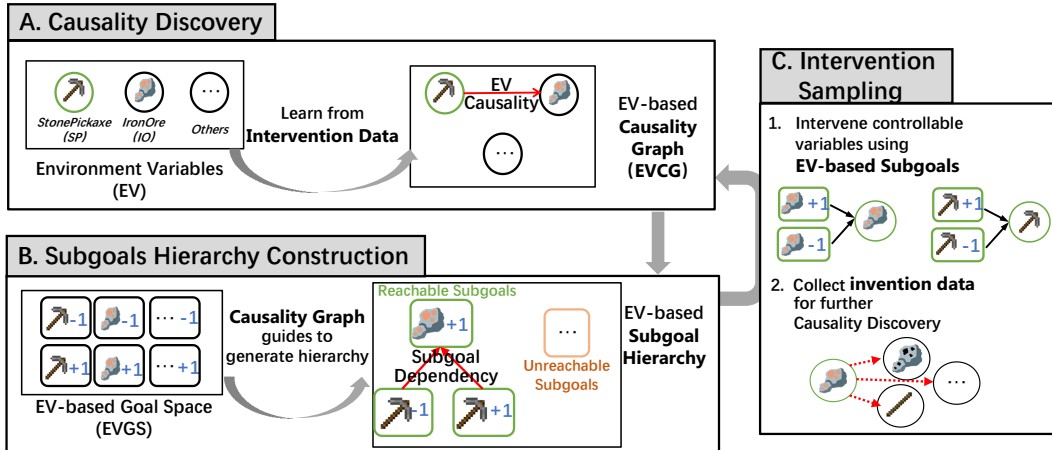

Figure 1: Overview of CDHRL framework. **A. Discover causality between Environment Variables (EVs):** Taking EV in Minecraft as a simplified example, we discover the causality between StonePickaxe (SP) and IronOre (IO) from SP's intervention data. **B. Build EV-based subgoal hierarchy for HRL subgoal training.** Guided by causality from SP to IO, we add reachable subgoals of increasing/decreasing IO values to the subgoal hierarchy and train corresponding subgoal-based policies. **C. Obtaining intervention data for causality discovery in the new round.** After getting subgoal-based policies about IO, the agent can further change IO's distribution and collect intervention data required in causality discovery (Step A) of the next iteration. (The pseudo-code of the framework is in Appendix B.1)

$c_{ij} = 1$ indicates that $X_j$ is the parent of $X_i$, which means that $X_j$ is the direct cause of $X_i$. $N_i$ is the jointly-independent noise and $Pa(i, C)$ is the collection of the parent nodes (direct cause variables) of $X_i$ in causality graph $C$.

**Intervention sampling** is a typical operation in causal discovery. Unlike standard sampling, it sets the distribution of certain variables in the system as fix value or uniform distribution, then samples to get intervention data. Intervention data has more causal information than observation data. In RL environments, we cannot do intervention directly because the variable generation mechanisms in the environment are unavailable. However, since the agent's behavior can affect the state of the environment, we train subgoal-based policies in CDHRL that can increase or decrease the values of variables to conduct the intervention.

## 4 Causality-Driven Hierarchical Reinforcement Learning

In this section, we present our Causality-Driven Hierarchical Reinforcement Learning (CDHRL) framework that leverages the causality of Environment Variables for high-efficient hierarchical structure discovery. Figure 1 shows the architecture of CDHRL. To enable the causality discovery and the subgoal hierarchy construction to promote each other, we first adopt environment variables to build a causality-based hierarchical structure that bridges causality discovery and hierarchical reinforcement learning (Section 4.1). Then, we provide the causality discovery methodology in the context of MDP (Section 4.2 & Section 4.4). Additionally, we propose the subgoal policy training methodology based on the causality-based hierarchical structure (Section 4.3).

### 4.1 Causality-based Hierarchical Structure

To bridge causality discovery and subgoal hierarchy construction, and to make them collaboratively promote each other, we introduce the main components of these two processes, causality graph and goal space, both based on environment variables. The **Environment Variables** (EV) are disentangled factors of the environment observation, which can include noise or lack some key information of the true environment state. In other words, EV does not mean the perfect abstract representation of the environment state that the experts provide. Obtaining EV is relatively convenient for environments providing state vector-based observation since information of different dimensions is disentangled.

For environments with purely image-based observation, finding EV has been largely studied, like CausalVAE [9] and DEAR [10]. Environment Eden and 2d-Minecraft both offer state vector-based observation. Considering that CDHRL focuses on the hierarchical structure discovery guided by the causality, we assume that EV can be obtained from observation by a function, $O : \mathcal{S} \to \mathcal{X}$, which intercepts part of observation as EV vector. We also clarify EV in detail and conduct a sensitivity analysis on EV in Appendix A to verify the reasonableness of the function $O$.

**Environment Variable-based Causality Graph (EVCG).** The causality graph describes the system's causality through a directed acyclic graph (DAG), where the source nodes of directed edges represent cause variables and destination nodes represent effect variables. The causality graph is shown in Figure 1.A. The process learns the environment variable-based causality graph (EVCG) of the RL environment, where nodes consist of environment variables and edges represent causality. Environment Variables are key factors of environment state and the fundamental element of causality. Taking Minecraft as an example, environment variables consist of the number of tools (like StonePickaxe), materials (like IronOre), and surrounding objects (like Pigs). The causality represented by the arrow from StonePickaxe to IronOre means that the number of stone pickaxes the agent-owned affects the probability of acquiring iron ore.

**Environment Variable-based Goal Space (EVGS).** To leverage the discovered causality between the environment variables, we introduce environment variable-based goal space (EVGS) and construct the subgoal hierarchy upon EVGS, as shown in Figure 1.B. Different from prior studies that adopt the state or latent spaces as subgoal space [19, 25], EVGS consists of goals which are to change the value of environment variables. Thus the subgoal-based policies can be used to change the distribution of environment variables or complete downstream tasks. Formally, we introduce two fundamental changes, Increase ($F_i$) and Decrease ($F_d$), for variable values:

$$
\begin{aligned}
F_i(x_{i,t}, x_{i,t+1}) &= \mathbf{1}_{x_{i,t+1} > x_{i,t}} \\
F_d(x_{i,t}, x_{i,t+1}) &= \mathbf{1}_{x_{i,t+1} < x_{i,t}}.
\end{aligned}
\tag{2}
$$

where $X_{i;t}$ is the value of $X_i$ in state $s_t$. The EVGS is the Cartesian product of the environment variable set $\mathcal{X}$ and the changes set $\mathcal{F}_{change} = \{F_i, F_d\}$:

$$
\mathcal{G}_{EVGS} = \{(X_i, F)|X_i \in \mathcal{X}, F \in \mathcal{F}_{change}\}.
\tag{3}
$$

The corresponding goal-reaching function $r(s_t, g, a, s_{t+1})$ is:

$$
r(s_t, (X_i, F), a, s_{t+1}) = F(O(s_t)_i, O(s_{t+1})_i),
\tag{4}
$$

where $O$ is the mapping from state to variable values.

**Collaboration between Causality Discovery and Subgoal Hierarchy Construction.** With the EV-based causality graph and EV-based goal space, we can naturally transform the causality graph into a subgoal hierarchy. The subgoal hierarchy contains reachable subgoals selected from $\mathcal{G}_{EVGS}$ through the causality graph and are organized by the causality between variables. Step A and B in Figure 1 show an example. Specifically, we divide the environment variables into *controllable* and unknown ones for the agent. The controllable variables are variables whose related subgoals have been added into the subgoal hierarchy (green ones, e.g., StonePickaxe) so that the agent can change their distribution to obtain their intervention data. Hence, in each iteration, the agent can discover causality from controllable variables (e.g., StonePickaxe) to the unknown variables (e.g., IronOre). Then, some unknown variables, which as the effect of the causality, are considered as new controllable variables for the agent, because the agent's behavior can affect them and thus recognize their cause variables. The subgoals related to new controllable environment variables are considered to be *reachable*. We then train reachable subgoal-based policies and add them to the subgoal hierarchy. In the next iteration, the agent can change the distribution of new controllable environment variables through subgoal-based policies and discover new causality which takes them as the cause variable. The iteration converges when there is no new causality, and the remained isolated variables are uncontrollable for the agent. Subgoals of uncontrollable variables will not be added to the subgoal hierarchy.

In this way, causality discovery and subgoal hierarchy construction can promote each other. On the one hand, the discovered causality between environment variables guides the agent to find available subgoals accurately and construct the subgoal hierarchy upon EVGS. On the other hand, the agent can utilize trained policies of subgoals to do intervention on environment variables. More complex causality between variables can be discovered along with the sampling data from poor to rich. The alternating iterative optimization above can address the challenge of discovering hierarchical structure.

## 4.2 Causality Learning

**Causality in the Context of Markov Decision Process (MDP).** We first introduce two key points of causality discovery in the context of MDP: what causality to discover and how to get learning data. 1) *Causality is discovered within adjacent steps.* According to the properties of MDP, the state $s_{t+1}$ is only determined by state $s_t$ and action $a_t$. Hence, we redefine the causality equation 1 as $X_{i,t+1} := f_i(X_{pa(i,C),t}, N_i)$, which means the variable values at step $t+1$ are only affected by that at step $t$. In other words, we aim to learn the causality from transition data of adjacent steps in the agent's trajectory. 2) *Intervention data is obtained through subgoal-based policy.* As mentioned in the preliminary, it is hard to do intervention in the RL environment directly since the generation mechanisms are unavailable. Therefore, we control the variables' distribution through the agent's behavior to sample intervention data for causal discovery. Specifically, with the application of subgoals in EVGS, the agent can change the distribution of controllable environment variables and collect the intervention data subsequently. More details about the intervention process based on subgoal-based policy are introduced in Section 4.4.

**Structural Causal Model Learning.** The structural Causal Model (SCM) is a kind of model that describes the system's causality. We adopt a causal discovery methodology similarly to SDI [16] and intervention data collected from the agent's trajectories to learn the environment variables' SCM. As the equation 1 shown, SCM over $M$ variables can be described by two sets of parameters: **structural parameters** $\eta \in R^{M \times M}$ model the causality graph $C$ and **functional parameters** $\theta s$ model the generating functions $f$.

The parameters $\eta$ and $\theta$ are optimized through a two-phase iterative and continuous process. In the first function learning phase, we want to fix the input variables of generating function and optimize the functional parameters $\theta$. So we sample hypothesis configuration $C$ of the causality graph from Bernoulli distribution $Ber(\sigma(\eta))$, where $\sigma$ represents the Sigmoid function. And then, we optimize $\theta$ by maximizing the likelihood of the intervention data. In the second structure learning phase, we want to learn the causality graph under fixed generating functions $f$. We estimate the gradient of $\eta$ through fixed $\theta$ and a REINFORCE-like predictor proposed by Bengio et al. [3]. For the causality from variable $X_j$ to $X_i$, the gradient $g_{ij}$ of $\eta_{ij}$ is estimated from $X_j$'s intervention data set $\{X\}$ by the following formula:

$$g_{ij} = \sum_k (\sigma(\eta_{ij}) - c_{ij}^{(k)}) \left( \frac{L_{C,i}^{(k)}(X)}{\sum_{k'} L_{C,i}^{(k')}(X)} \right), \forall i \in \{0, \ldots, M-1\}, \tag{5}$$

where $k$ represents the k-th draw of hypothesis configuration $C$. $L_{C,i}^{(k)}(X)$ represents the $X_i$'s likelihood under the intervention data. The two optimization phases cycle until convergence. (Algorithm details are in Appendix B.2).

## 4.3 Subgoal-based Policy Training

As described in section 4.1, when new causalities are discovered, the subgoals related to new effect variables in the EVGS are added to the subgoal hierarchy by the guidance of new causalities. Figure 1.B shows the effect variable IO's subgoals are added to the subgoal hierarchy. Specifically, we first assumed the new effect variable $X_i$ is controllable, and subgoals of $X_i$ are reachable. We then train $X_i$'s subgoal-based policies $\pi(a|s, (X_i, F))$. To induce an efficient training, the action space of $\pi(a|s, (X_i, F))$ consists of subgoals related to $X_i$'s cause variables and primitive action set $\mathcal{A}$:

$$a \in \{(X_j, F')|X_j \in X_{pa(i,C)}, F' \in \mathcal{F}_{change}\} \bigcup \mathcal{A}. \tag{6}$$

Besides, to verify whether newly associated variables are controllable as supposed, we compare the final training success rates of the new subgoals with a preset threshold $\phi_{causal}$ before adding them to the subgoal hierarchy. When the corresponding subgoal-based policies are successfully trained, variables will be confirmed as controllable variables and can be intervened to obtain data in the next iteration. Along with the causality discovered from shallow to deep, the subgoal-based policies are trained layer by layer and finally consist of a multi-level hierarchy. After the above process converges to no new causality, subgoals related to remaining isolated variables are taken as unreachable ones. For details on subgoal-based policy training, please refer to the training algorithm in Appendix B.3.

Distinguishing reachable subgoals related to controllable environment variables and filtering unreachable ones during policy training can reduce the goal space reasonably. Meanwhile, through

the dependency of subgoals suggested by causality between environment variables, we construct the action space for the subgoal-based policies, which is more efficient than primitive action space. The above two improvements in policy training lead to a more data-efficient learning process, verified in experiment results in Section 6.2).

### 4.4 Intervention Sampling

Considering that environment variables in RL environments cannot be directly intervened, we propose a methodology that adopts subgoal-based policies to obtain the intervention data. Benefit from variable-based goal space, the subgoal-based policy could effectively change the distribution of controllable variables.

The detailed intervention sampling includes the following steps: **(1)** For controllable variable $X_i$, sample the desired value $x_i$ from the uniform distribution, and then use subgoals $(X_i, F)$ to drive the agent to change the value of $X_i$ to $x_i$; **(2)** Collect the follow-up trajectory until $X_i$'s value changes, and then obtain intervention data of $X_i$ from all variables' values of adjacent steps. At the beginning of the optimization, there are no trained subgoal-based policies. Therefore, we set the agent's action as one of the variables first and do intervention on it to initiate the iterative process.

## 5   Experiment Setup

**Environment**   To verify the effectiveness of CDHRL, we choose two environments with discrete action space, namely 2d-Minecraft [30] and Eden [4] (more details of environments are introduced in Appendix C.1. They have complex hierarchical structures and very sparse rewards. **2D-Minecraft** [30] is a simplified 2D version of the famous Minecraft [14]. In one episode with limited steps, the agent must navigate the map, pick up various materials, and craft tools until obtaining the diamond. The agent only can get a positive reward after getting the diamond. **Eden** [4] is a survival game which is similar to "Don't Starve." To make a living in a $40 \times 40$ grid world, the agent with $8 \times 8$ limited vision range must chase animals to obtain food for maintaining satiety, find rivers to get water for preventing being thirsty, and collect materials to craft tools for surviving in the night. The agent takes survival as the unique target and cannot get any positive rewards except a negative reward after it dies and ends the episode.

**Implementation**   We implement the agent based on multi-level DQN with HER [1], considering the action space is discrete. The training is divided into two stages, pre-training and adaptation. In the pre-training stage, we train multi-level subgoal-based policies until no new causality is discovered. In the adaptation stage, we train an upper controller on the pre-trained subgoals to maximize the task reward. The SCM is implemented like SDI [16]. The environment supports the function $O$ from environment state to variable values.

**Baselines**   We compare CDHRL with the following three methods for validation. For a fair comparison, we use the environment variable-based goal space $\mathcal{G}_{EVGS}$ to replace the goal space in these algorithms to ensure that the performance promotion is because of the difference in hierarchical structure discovery methods. **Oracle HRL (OHRL)** is implemented as a two-level DQN with HER and an *oracle goal space*. The goal space is a subset of $\mathcal{G}_{EVGS}$ after artificial eliminating unreachable and useless subgoals. **HAC** [19] is a powerful goal-conditioned HRL that discovers subgoals with a randomness-driven exploration paradigm. We implement a two-layer HAC **with subgoal space** $\mathcal{G}_{EVGS}$. **LESSON** [20] is a modified goal-conditioned HRL method based on HAC that discovers subgoals from slowly changed features. **MEGA** [27] is a kind of goal-conditioned HRL enhanced by curriculum learning, which pre-train subgoals in the order of their training progress. We set the initial subgoal distribution as the uniform distribution on $\mathcal{G}_{EVGS}$ and pre-trains subgoals with enough steps. After that, like the adaption stage in CDHRL, we train an upper controller on the trained subgoals.

## 6   Results

We first carry out extensive experiments to compare the performance of CDHRL and the state-of-the-arts in complex environments with a hard-to-explore hierarchical structure. Then, we dig deep ablation studies to show why CDHRL triumphs over the existing HRL paradigm. At last, we showcase the discovered causality-based hierarchical structure, which can be reasonably interpreted.

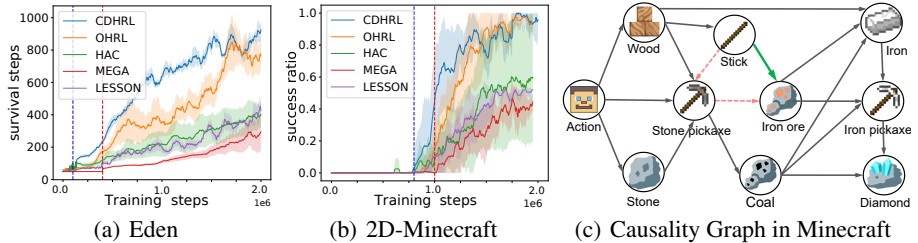

| (a) Eden | (b) 2D-Minecraft | (c) Causality Graph in Minecraft |

Figure 2: (a) Agent's survival time in Eden. (b) Success ratio in 2D-Minecraft. The vertical dotted lines indicate the end of pre-training of CDHRL and MEGA. Results are derived from average data in 8 trials. (c) The causal graph of 2D-Minecraft discovered by the agent. Some uncontrollable variables that unlinked are ignored here.

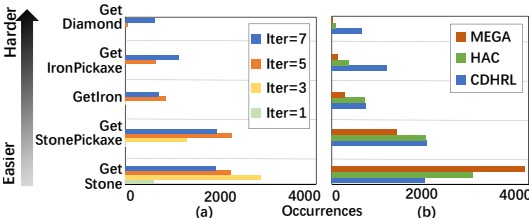

Figure 3: Exploration capability comparison. We select five explore milestones in 2D-Minecraft from easy-to-explore to hard-to-explore and record the occurrences in 10K test episodes to compare the agent's exploration capability. More occurrences on the hard-to-explore milestone represent higher exploration capability. (a) CDHRL's exploration capability iteratively increases along with the construction of hierarchical structures. (b) CDHRL shows much better exploration capability than HAC and MEGA. All compared methods are tested after trained 800K steps.

## 6.1 Comparative Analysis

We compare the performance on Eden and 2D-Minecraft of CDHRL with other methods. Figure 2 shows that CDHRL learns faster and achieves the best performance even though all the evaluated methods have the same subgoal space. Concretely, OHRL, HAC and LESSON do not have the pre-training stage and start learning directly. However, the success ratios of OHRL, HAC and LESSON in the first 1e6 steps of 2(b) keep nearly zero since they seldomly obtain the success trajectory data of getting diamonds. CDHRL learns much faster even if there is pre-training time overhead. MEGA has the pre-training stage for curriculum learning to identify the near-term and long-term subgoals. However, it fails to identify the effective hierarchical structures in these two environments with relatively large sub-goal space. The hierarchical structure discovered by CDHRL not only significantly enhances the agent's exploration capability in sparse reward, but also identifies the reachable subgoals that promotes the learning speed of subgoal-based policies. We will explain these phenomenons further in Section 6.2.

## 6.2 Insights

Further, we evaluate the detailed exploration capability and causal discovery efficiency and show how CDHRL outperforms existing HRL paradigms. We have two observations from the experiment results:
**1) The causality-driven exploration paradigm enhances the exploration capability significantly.**
We experimentally evaluate the agent's exploration capability in 2D-Minecraft based on the achieved times of exploration milestones in the environment. Figure 3(b) shows the exploration capability comparison of different methods. The CDHRL's achieved times of hard-to-explore milestones are much higher than HAC and MEGA, indicating a much better exploration capability of CDHRL. Figure 3(a) shows the exploration capability increases along with iterations of causality discovery in CDHRL. The reason is that the agent's causality-driven exploration becomes more efficient as more subgoal-based policies and controllable variables, which other HRL methods cannot achieve. In conclusion, with the help of the causality-based exploration paradigm, the agent can achieve better exploration efficiency.

**2) The subgoal hierarchy construction and causality discovery iteratively promote each other.**
On the one hand, the hierarchical structure makes the causality graph more accurate. We train the

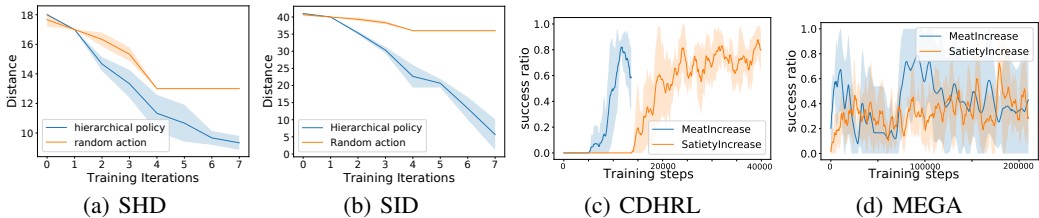

| (a) SHD | (b) SID | (c) CDHRL | (d) MEGA |

Figure 4: (a) Structure Hamming Distance (SHD) and (b) Structural Interventional Distance (SID) between learned and ground-truth causality graph. The intervention data sampled by the assistance of hierarchical policy make the causality graph more accurate. The ground-truth causality graphs of Eden and 2D-Minecraft are in Appendix C.2.(c) and (d) are learning curve of two subgoals.

causal model in 2D-Minecraft using intervention data collected by hierarchical policy and random action, respectively. We record the causality graph of each iteration and compute their *Structure Hamming Distance* (SHD) and *Structural Interventional Distance* (SID) relative to the ground truth, where SHD and SID are typical metrics for the accuracy of causality graphs. The lower distance means the more accurate causality graph. Figure 4(a) and 4(b) show the result. The accuracy of the causality graph learned using the hierarchical policy is much better than using random action, especially for SID. Such results indicate that utilizing the hierarchical policy to obtain intervention data is essential for causality discovery.

On the other hand, the construction of the subgoal hierarchy is more efficient and reasonable. We compare the convergence speed and stableness of CDHRL and MEGA to achieve two subgoals in Eden. For these two subgoals, $MeatIncrease = (X_{Meat}, F_i)$ and $SatietyIncrese = (X_{Satiety}, F_i)$, the former is the prerequisite goal of the latter one and should be learnt by agent first. Figure 4(c) and 4(d) show that CDHRL achieves a faster and more stable learning process than MEGA. We attribute the poor performance of MEGA to the fact that it cannot identify the dependency of the subgoals and is unable to learn subgoals in order, resulting in much lower performance, especially when the environment is complex with a complicated dependency of subgoals. In contrast, since $X_{Meat}$ is the cause variable of $X_{Sariety}$, CDHRL infers the learning order of subgoals based on discovered causality and achieves stable and fast learning progress. Additionally, the reduced action space of multi-level policy transformed from causality also improves training efficiency.

### 6.3 An Showcase of Causality-based Hierarchical Structure

We showcase the established hierarchical structure based on discovered causality in 2D-MineCraft from the causality graph view. The learned causality graph can be reasonably interpreted. As shown in Figure 2(c), most of the causality in the graph meets with human cognition. We also find an interesting phenomenon that CDHRL sometimes may discover long-term causality and ignore corresponding short-term causality, e.g. "Stick" to "Iron re" (green edge) instead of "Stick" to "Stone pickaxe" to "Iron ore" (red edges) in Figure 2(c).CDHRL may benefit from such phenomenons because cutting unimportant causality details can improve the training speed and the generalization of hierarchical structure. Although there is some deviation between the discovered and the ground-truth causality graph, the causality-based hierarchical structure still improves the exploration efficiency.

## 7 Conclusion

Hierarchical reinforcement learning methods can hardly work well in complicated environments because of the low exploration efficiency of the randomness-driven exploration paradigm when discovering the hierarchical structure. We propose a causality-driven hierarchical reinforcement learning framework (CDHRL) to address the challenge, which autonomously builds high-quality hierarchical structures in an iterative boosting way. In two complicated environments with very sparse rewards, 2d-Minecraft and Eden, our method discovers a high-quality subgoal hierarchy and significantly enhances exploration efficiency. Limited in causal discovery algorithm capability and the complexity of variable changes set, our method can only be applied in environments with discrete environment variables. Moreover, CDHRL needs an extra disentangled encoder in purely image-based environments for finding environment variables. We leave broadening the application scope of CDHRL as our future work.

## Acknowledgements

This work is partially supported by the National Key Research and Development Program of China(under Grant 2018AAA0103300), the NSF of China(under Grants 61925208, 62002338, 62102399, U19B2019, 61732020), Beijing Academy of Artificial Intelligence (BAAI), CAS Project for Young Scientists in Basic Research(YSBR-029), Youth Innovation Promotion Association CAS and Xplore Prize.

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
