# OpenReview forum: "Causality-driven Hierarchical Structure Discovery for Reinforcement Learning"
_NeurIPS.cc/2022/Conference — NeurIPS 2022 Accept_

### Official Review · Reviewer_vmvq · 2022-07-06

**Rating:** 6
**Confidence:** 3
**Soundness:** 2 fair
**Presentation:** 2 fair
**Contribution:** 3 good

**Summary:**

The authors propose a formulation of hierarchical reinforcement learning that utilizes causal structure between environmental variables to generate subgoals. The discovered subgoals and lower-level policies generated to reach the subgoals are subsequently adapted using a learnt hierarchical agent that solves downstream tasks. The authors test their method in addition to several hierarchical RL baselines on the 2D-Minecraft and Eden environments and obtain promising results on the both. They show faster convergence in both environments in addition to interpretable causal graphs.


**Questions:**

See the Weaknesses section for my questions. I believe adding incorporating your answers to the draft would be beneficial.

**Limitations:**

See the Weaknesses section for my analysis of the assumptions/limitations of the work. I don't believe there is any cause for concern for negative societal impact.

**Strengths And Weaknesses:**

Strengths:
1. The problem considered by the authors is one of strong relevance.
2. The approach is principled with well-made figures.
3. The results are repeated across seeds and are in challenging environments.

Weaknesses
1. Directionality: It's unclear how the algorithm learns directionality for causes and effects. From my understanding, the interventional data is generated by lower-level policies in the action space that attempt to set the environmental variables to a particular value. Thus, it appears the order in which environment variables are selected matters. What if a variable is selected in the initial iterations for which the causal parents are discovered in a later iteration? Is the causal graph updated in such a setting? For example, if the agent decides to learn policies that intervene on Ironore before it intervenes on stonepickax, what happens then?
2. Commentary on Scalability of during large number of EVs: I believe that the authors need to add some information about the scalability of their approach with environmental variables. In the current draft, they say “ to verify whether newly associated variables are controllable as supposed, we compare the final training success rates of the new subgoals with a preset threshold phi before adding them to the subgoal hierarchy”. How are they selected? How many are selected? How are the lower-level policies learned? I would imagine through equation (4), but how long do you train, etc should be added.
3. The algorithm should not be relegated to Supplementary work. This draft sorely misses the details afforded by it and would help ground a lot of the description.
4. Some assumptions on the density of the underlying causal graph: I believe the authors need to state some assumptions about the density of the underlying causal graph, in particular, that currently, they assume that the underlying causal graph is dense and that each of the environment variables influences the others. I can imagine a situation where mechanisms are sparse and that learning policies which intervene on a causal factor only allows one to intervene on the immediate children of the same causal factor, but these children no longer have any other effect variables linked to them.
5. Description of Subgoal Hierarchy needed: What is a subgoal hierarchy? From reading the paper several times, I was able to discern that is an ordering on the elements in the goal space that are linked due to their temporal dependencies, i.e., getting IO is a child of SP because SP is needed to mine IO. Some mathematical precision is necessary to describe this especially since it forms the core of your algorithm. Compressing section 5 by avoiding newlines will generate room enough for such an explanation.
6. Assumptions on the utility of causal graphs for subgoal: A central assumption made by the authors is that in their setup, nodes on the causal graph are all useful in goal generation. This is in general not true and worth stating. For example, manipulation by a robotic hand of an object in its palm does not require knowledge of friction coefficients between the block and the ground.
7. Curious that the oracle is outperformed by your method. Is this a bug? Should it not be that the oracle with ground-truth causal graph inputs beat discovery?
8. Missing citing work: https://arxiv.org/abs/2010.03110 and https://proceedings.neurips.cc/paper/2021/file/c1722a7941d61aad6e651a35b65a9c3e-Paper.pdf and https://ieeexplore.ieee.org/document/9561439

---

> ### Author Response · Authors · 2022-08-02
> **(Part 1/2) Response  to Reviewer vmvq**
>
> We thank the reviewer for the insightful comments and suggestions. We would like to clarify the concerns as follows:
>
> 1. > Weakness 1: Directionality: It's unclear how the algorithm learns directionality for causes and effects. From my understanding, the interventional data is generated by lower-level policies in the action space that attempt to set the environmental variables to a particular value. Thus, it appears the order in which environment variables are selected matters. What if a variable is selected in the initial iterations for which the causal parents are discovered in a later iteration? Is the causal graph updated in such a setting? For example, if the agent decides to learn policies that intervene on Ironore before it intervenes on stonepickax, what happens then?
>
>    We have considered the cold start of intervention sampling (see the description of intervention sampling, lines 271-273). In detail, we initially set the agent’s action as one of the variables and do intervention on it to initiate the iterative process. Such a strategy ensures that:
>    1. The initial variable is controllable;
>    2. The initial variable is the cause variable of other controllable variables in the environments;
>    3. The initial variable is not the effect variable of other controllable variables (since the agent self generates the action).
>
>    Besides these assurances, as described in the paper (lines 193-195), we discover causality from controllable variables to unknown variables in each round to avoid such a trap.
>
> 2. > Weakness 2: Commentary on Scalability of during large number of EVs: I believe that the authors need to add some information about the scalability of their approach with environmental variables. In the current draft, they say “ to verify whether newly associated variables are controllable as supposed, we compare the final training success rates of the new subgoals with a preset threshold phi before adding them to the subgoal hierarchy”. How are they selected? How many are selected? How are the lower-level policies learned? I would imagine through equation (4), but how long do you train, etc should be added.
>
>    The saying "to verify whether ... to the subgoal hierarchy" means that for each newly associated variable, we first assume it is controllable. And then, based on the variable change function set, we train two subgoal policies that increase and decrease the variable value. After training, only when the training success rates of the two subgoals are higher than a preset threshold can we ensure the corresponding variable is controllable, and the two subgoals can be added to the subgoal hierarchy.
>
> 3. > Weakness 3: The algorithm should not be relegated to Supplementary work. This draft sorely misses the details afforded by it and would help ground a lot of the description.
>
>    Thanks for your suggestion. Because of page constraints, there is no space for the pseudo-code in the main body. We have revised the description of the training process to make a better understanding.
>
> 4. > Weakness 4: Some assumptions on the density of the underlying causal graph: I believe the authors need to state some assumptions about the density of the underlying causal graph, in particular, that currently, they assume that the underlying causal graph is dense and that each of the environment variables influences the others. I can imagine a situation where mechanisms are sparse and that learning policies which intervene on a causal factor only allows one to intervene on the immediate children of the same causal factor, but these children no longer have any other effect variables linked to them.
>
>    In fact, we do not assume the density or connection form of the underlying causality graph. The hard-to-explore complex environment we considered means that the underlying causality graph should contain relatively many variables and long causality chains rather than dense connections. The longer the causality chains, the harder it is to reach the deep subgoals. Our method is designed to discover subgoals from shallow to deep. For example, the underlying causality graph of our experiment environment Eden is sparse connected (see the causality graph discovered in Eden https://anonymous.4open.science/r/nips2022_id5444_CDHRL_rebuttal/Eden_cg.png). The sparse connection is actually an assumption of all causality graph-based methods.

---

> > ### Author Response · Authors · 2022-08-02
> > **(Part 2/2) Response to Reviewer vmvq**
> >
> > 5. > Weakness 5: Description of Subgoal Hierarchy needed: What is a subgoal hierarchy? From reading the paper several times, I was able to discern that is an ordering on the elements in the goal space that are linked due to their temporal dependencies, i.e., getting IO is a child of SP because SP is needed to mine IO. Some mathematical precision is necessary to describe this especially since it forms the core of your algorithm. Compressing section 5 by avoiding newlines will generate room enough for such an explanation.
> >
> >    Thanks for your suggestion. We have added a more clear description of the subgoal hierarchy in the method section  (see lines 186-190).
> >
> > 6. > Weakness 6: Assumptions on the utility of causal graphs for subgoal: A central assumption made by the authors is that in their setup, nodes on the causal graph are all useful in goal generation. This is in general not true and worth stating. For example, manipulation by a robotic hand of an object in its palm does not require knowledge of friction coefficients between the block and the ground.
> >
> >    We explain the assumption and the example from two aspects.
> >
> >    A. Actually, we do not mean “nodes on the causal graph are all useful in goal generation” but “nodes on the causal graph are all controllable .” The environment variable-base subgoal space (EVGS) is the hypothesis space. Our method discovers reachable subgoals in EVGS (since our method discovers reachable subgoals in an unsupervised way and cannot confirm what subgoals are useful for downstream tasks) and dependencies between them. In practice, our environment variable set comes from state vector-based observation and contains noise or uncontrollable variables. These variables would not occur in the causality graph, and the corresponding subgoals would not participate in the subgoal hierarchy (we describe this automatic filtering process in section 4.3).
> >
> >    B. Besides, we construct the potential action space for subgoal policies and downstream tasks. For subgoal policies, the action space consists of primitive actions of the agent and cause subgoals. For downstream tasks, the action space consists of all reachable subgoals. For example, considering manipulation by a robotic hand, if the friction is not the cause of manipulation, or the manipulation is simple and needs not to consider friction, then the policy would not leverage the subgoals about friction but the primitive actions or other subgoals.
> >
> > 7. > Weakness 7: Curious that the oracle is outperformed by your method. Is this a bug? Should it not be that the oracle with ground-truth causal graph inputs beat discovery?
> >
> >    The Oracle HRL refers to a two-level hierarchical structure with human-selected useful subgoals. As we described in section 6.1 (lines 295-297), compared with CDHRL, the two-level structure lacks dependencies between subgoals, which makes its subgoal training efficiency lower than CDHRL. But they reach similar converge performance.
> >
> > 8. > Weakness 8:
> >
> >    Thanks for your suggestions; we will add citations of these papers in the related works.

---

> > > ### Comment · Reviewer_vmvq · 2022-08-09
> > > **Thanks**
> > >
> > > Thank you for your rebuttal. I will keep my score for the paper.

---

### Official Review · Reviewer_K7mE · 2022-07-11

**Rating:** 6
**Confidence:** 3
**Soundness:** 3 good
**Presentation:** 3 good
**Contribution:** 3 good

**Summary:**

This paper proposes a new hierarchical reinforcement learning method that could automatically discover hierarchical structure with a learned causality graph. To build the causality graph, the proposed method introduces the “Environment Variables” as nodes in that graph, which requires some task-specific knowledge. The causality graph and subgoal-based hierarchical policy are simultaneously learned and facilitate each other.

**Questions:**

The assumption of oracle Environment Variables is a little strong. Could the proposed method generalize to more common settings where no oracle Environment Variables are available?

The blue learning curve in Figure 4(c) has not been plotted to 40000 training steps. Will it converge steadily at the end of the learning?


**Limitations:**

The authors have discussed the limited application to environments with discrete environment variables in Section 7, and have not touched on another limitation of requiring oracle environment variables. Those two limitations are related.

**Strengths And Weaknesses:**

## Significance
Automatic hierarchical structure discovery is a fundamental research problem in deep reinforcement learning. This paper provides a new way to solve this problem, which owes remarkable significance.

## Originality
The idea of using a causality graph to discover hierarchical structure is novel. However, the proposed method relies on the “Environment Variables”. The way of obtaining the “Environment Variables” is non-trivial, so I doubt whether this method could be applied to more general settings, where the oracle “Environment Variables” are hardly accessible.

## Clarity
The writing of this paper is mostly clear, but seems in a rush. For example, there is a latex compile error in Line 485. The authors are encouraged to polish up the writing in the future.

## Quality
The idea of using a causality graph to facilitate the discovery of hierarchical structure makes sense, since there are causality relationships among subtasks in most hierarchical tasks. The proposed method is compared against previous goal-conditioned HRL works. However, claiming HAC as a state-of-the-art method is not precise. A lot of more advanced goal-conditioned HRL works have emerged in recent years [1, 2], where the World Model paper also proposed to build a graph to guide the subgoal selection. The authors are encouraged to compare the proposed method to more advanced goal-conditioned HRL methods.

[1] Zhang er al., World Model as a Graph: Learning Latent Landmarks for Planning, ICML 2021.
[2] Li et al., Learning Subgoal Representations with Slow Dynamics, ICLR 2021

---

> ### Author Response · Authors · 2022-08-02
> **(Part 1/2) Response to Reviewer K7mE**
>
> We thank the reviewer for the insightful comments and suggestions. We would like to clarify the concerns as follows:
>
> 1. > Originality:
>    > The idea of using a causality graph to discover hierarchical structure is novel. However, the proposed method relies on the “Environment Variables”. The way of obtaining the “Environment Variables” is non-trivial, so I doubt whether this method could be applied to more general settings, where the oracle “Environment Variables” are hardly accessible.
>
>    For environments providing state vector-based observation (which cover a broad category of environments, such as Atari [3], Mujoco [4], 2d-Minecraft [5], ...), obtaining "Environment Variables" (EV) is relatively convenient. For CDHRL, EV are disentangled factors in the environment observation rather than the perfect abstract representation states that experts provide. The state vector-based observation can be directly transformed into the EV vector since information of different dimensions is disentangled (In our experiment, we take disentangled factors in observation as the EV, including many noise and useless variables). Many HRL methods [2, 5, 6, 7, 8] commonly use them as the standard inputs. For example, LESSON [2] assumes that each input state dimension represents an independent feature. DSC [7] uses disentangled factors like position, orientation, linear velocity, rotational velocity, and a has_key indicator as the state space. For environments with image-based observation, finding EV has been largely studied, like CausalVAE [9] and DEAR [10]. Considering that our paper focuses on how to build the subgoal hierarchy instead of finding EV, we do not include that part in this paper. Thanks for the great advice, and we will integrate the EV exploring in our CDHRL paradigm for environments with image-based observations in the future.
>
>    To note, we also perform sensitivity analysis on EVs (see Appendix A), and the results show that noises or confused EVs would not significantly affect CDHRL's effectiveness.
>
>    > [1] Zhang et al., World Model as a Graph: Learning Latent Landmarks for Planning, ICML 2021.
>    >
>    > [2] Li et al., Learning Subgoal Representations with Slow Dynamics, ICLR 2021
>    >
>    > [3] Anand et al., Unsupervised State Representation Learning in Atari.  NeurIPS 2019
>    >
>    > [4] Todorov et al., MuJoCo: A physics engine for model-based control.
>    >
>    > [5] Sohn et al., Hierarchical Reinforcement Learning for Zero-shot Generalization with Subtask Dependencies. NeurIPS 2018
>    >
>    > [6] Corcoll et al., DISENTANGLING CAUSAL EFFECTS FOR HIERARCHICAL REINFORCEMENT LEARNING
>    >
>    > [7] Bagaria et al., OPTION DISCOVERY USING DEEP SKILL CHAINING, ICLR 2020
>    >
>    > [8] Chuck et al., Hypothesis-Driven Skill Discovery for Hierarchical Deep Reinforcement Learning, IROS 2020
>    >
>    > [9] Yang et al., Causalvae: Disentangled representation learning via neural structural causal models. CVPR, 2021
>    >
>    > [10] Shen et al., Disentangled generative causal representation learning
>
> 2. > Clarity:
>    > The writing of this paper is mostly clear, but seems in a rush. For example, there is a latex compile error in Line 485. The authors are encouraged to polish up the writing in the future.
>
>    Thanks for your suggestions. We have revised our writings (Line 485 is the checklist example; we have deleted it).

---

> > ### Author Response · Authors · 2022-08-02
> > **(Part 2/2) Response  to Reviewer K7mE**
> >
> > 3. > Quality:
> >    > The idea of using a causality graph to facilitate the discovery of hierarchical structure makes sense, since there are causality relationships among subtasks in most hierarchical tasks. The proposed method is compared against previous goal-conditioned HRL works. However, claiming HAC as a state-of-the-art method is not precise. A lot of more advanced goal-conditioned HRL works have emerged in recent years [1, 2], where the World Model paper also proposed to build a graph to guide the subgoal selection. The authors are encouraged to compare the proposed method to more advanced goal-conditioned HRL methods.
> >
> >    A. LESSON[2]: We have added a new comparison with LESSON in the revised paper. Results show that LESSON performs similarly compared with HAC, while CDHRL outperforms them. LESSON aims to select slowly changed features in the state as subgoals. Such a strategy needs trajectories that reach subgoals; however, it cannot meet in hard-to-explore environments (LESSON is verified in the easy-to-explore ant-maze environment [2]). LESSON performs similarly to HAC because HAC and LESSON both utilize the strategy of hindsight goal relabeling.
> >
> >     (see https://ibb.co/YtgPxts (Eden) and https://ibb.co/CHMLVck (2d-Minecraft))
> >
> >    B. World Model (WMG)[1]: Based on model-based RL, WMG focuses on how to leverage graphs for planning instead of exploration, which has a quite different design focus from CDHRL. WMG learns a small number of landmarks in the goal space to construct a graph for planning, which implicitly assumes that the goal space exploration is easy. In summary, WMG and CDHRL target entirely different scenarios.
> >
> >    [1] Zhang et al., World Model as a Graph: Learning Latent Landmarks for Planning, ICML 2021.
> >
> >    [2] Li et al., Learning Subgoal Representations with Slow Dynamics, ICLR 2021
> >
> >
> >
> > 4. > Question 1: The assumption of oracle Environment Variables is a little strong. Could the proposed method generalize to more common settings where no oracle Environment Variables are available?
> >
> >    A similar question has been responded to (see 1. Originality). In conclusion, the proposed method CDHRL can easily generalize to environments with state vector-based observation.
> >
> >
> >
> > 5. > Question 2: The blue learning curve in Figure 4(c) has not been plotted to 40000 training steps. Will it converge steadily at the end of the learning?
> >
> >    As we described in the paper, CDHRL learns a multi-level policy. The subgoal (called A temporarily) represented by the blue curve is at a lower level than the subgoal (called B temporarily) represented by the red curve. Subgoal A trains before B, and after 40000 training steps, A keeps fixed to act as an element of B's action space when training B. This is because subgoal A's variable is the cause variable of B's variable in the discovered causality.
> >
> >
> >
> > 6. > Limitations: The authors have discussed the limited application to environments with discrete environment variables in Section 7, and have not touched on another limitation of requiring oracle environment variables. Those two limitations are related.
> >
> >    Thanks for your suggestions. CDHRL needs an extra disentangled encoder in the pure image-based environment. We have added this limitation to the Conclusion of the paper.

---

### Official Review · Reviewer_6RRy · 2022-07-11

**Rating:** 6
**Confidence:** 3
**Soundness:** 3 good
**Presentation:** 3 good
**Contribution:** 2 fair

**Summary:**

The paper proposes a causality driven hierarchical reinforcement learning framework to build hierarchical structures in an iterative boosting way. The paper shows experimental results in 2D-Minecraft and Eden environments with sparse rewards.

**Questions:**

It is not clear to me the need of using ideas from causality? Given some environment variables, relationships between variables can be discovered without ideas from causality.

Also, intervention sampling seems very close to the notion of exploration. Just that it is a bit smarter to explore in a smaller space using environment variables.

**Limitations:**

The assumptions that environment variables are given to us is the main limitation of the paper. Finding these environment variables along with the structure, is difficult.

**Strengths And Weaknesses:**

Strengths:
The paper is well written and clear. The experiments are well done.

Weakness:
The weakest part of the paper is that it is not clear where the novelty of the paper lies. It would be greatly appreciated if the authors differentiate their work against prior work.

Multiple previous work tries to find graph like structures in environments for finding skills/options etc
For example, Robot learning from demonstration by constructing skill trees.


Important Prior Work on HRL and Minecraft:
1) Forgetful Experience Replay in Hierarchical Reinforcement Learning from Demonstrations
2) Align-RUDDER: Learning From Few Demonstrations by Reward Redistribution

---

> ### Author Response · Authors · 2022-08-02
> **(Part 1/2) Response to Reviewer 6RRy**
>
> Thanks for the thoughtful comments. We would like to clarify the concerns as follows:
>
> 1. > Weakness: The weakest part of the paper is that it is not clear where the novelty of the paper lies. It would be greatly appreciated if the authors differentiate their work against prior work.
>    >
>    > Multiple previous work tries to find graph like structures in environments for finding skills/options etc For example, Robot learning from demonstration by constructing skill trees. Important Prior Work on HRL and Minecraft: ForgER [1] and Align-RUDDER [2]
>
>    Leveraging the graph structure for finding skills is the common design philosophy for HRL techniques. The key question is how to automatically construct the graph structure during the early stage without success trajectory. The novelty of our method (CDHRL) is that we introduce the causality-driven paradigm for subgoal hierarchy discovery. CDHRL can automatically find high-quality subgoals and dependencies among them in hard-to-explore and sparse reward environments from scratch. Specifically, CDHRL introduces the causality-driven paradigm that makes the agent automatically discover causality from shallow to deep, and further constructs the subgoal hierarchy from easy to hard. Such discovery cannot be achieved by the random-driven paradigm of existing HRL methods.
>
>    The ForgER [1] and Align-RUDDER [2] are imitation learning methods that leverage the graph discovered manually (hard-encoded way) from successful expert demonstrations to promote the data efficiency of imitation. Compared to them, CDHRL has the following advantages:
>
>    1. CDHRL efficiently trains subgoal policies from scratch without the successful expert demonstrations that are the premise of the imitation learning-based methods (such as ForgER [1] and Align-RUDDER [2]).
>    2. CDHRL automatically discovers the subgoal hierarchy in hard-to-explore environments, while ForgER [1] and Align-RUDDER [2]  manually identify the subgoal chain.
>
>    > [1] Skrynnik et al., Forgetful Experience Replay in Hierarchical Reinforcement Learning from Demonstrations
>    >
>    > [2] Patil et al., Align-RUDDER: Learning From Few Demonstrations by Reward Redistribution
>
> 2. > Question 1: It is not clear to me the need of using ideas from causality? Given some environment variables, relationships between variables can be discovered without ideas from causality.
>
>    The importance of causality has been described in the response to Weakness. The difference between "relationship" and "causality" is significant and is required for CDHRL.
>
>    The "relationship" (without any extra definition) is a kind of correlation without direction. For a famous example, rooster crowing and the sun rising are strongly correlated, but the rooster does not cause the sun to rise. So correlation cannot guide the agent to discover subgoals from cause to effect, in other words, from easy-to-reach to hard-to-reach. Only causality can guide the discovery of high-quality subgoals in hard-to-explore environments.
>
> 3. > Question 2: Also, intervention sampling seems very close to the notion of exploration. Just that it is a bit smarter to explore in a smaller space using environment variables.
>
>    Intervention sampling that leverages subgoal policies is the exploration part of our causality-driven paradigm. It bridges the causality discovery and subgoal hierarchy construction and makes them work in an iterative boosting way. "Intervention sampling" is very different from "exploration" in the random-driven paradigm of the existing HRL methods,  which implicitly explore through random action or random sub-policies. And Intervention sampling achieves "smarter to explore in a smaller space", which is exactly our desired results that can address high-quality subgoals discovery in hard-to-explore environments.

---

> > ### Author Response · Authors · 2022-08-02
> > **(Part 2/2) Response to Reviewer 6RRy**
> >
> > 4. > Limitation: The assumptions that environment variables are given to us is the main limitation of the paper. Finding these environment variables along with the structure, is difficult.
> >
> >    There may be some confusion about the environment variables, and we made clarification and revisions in the paper (The "oracle" does not mean "perfect"). We do not need EVs to be the perfect abstract representation states that experts provide. They are just disentangled factors in the environment observation. State vector-based observation is naturally disentangled, which is commonly adopted as the standard input in many HRL methods [3, 4, 5, 6, 7]. The state vector-based observation can be directly transformed into the EV vector since information of different dimensions is disentangled. In our experiment, we take disentangled factors in observation as the EV, including many noise and useless variables. Specifically, in 2d-Minecraft, we use the backpack contents as EV, which is also used in other RL methods for Minecraft [1, 2, 6] (these methods use the contents of the backpack to build sub-task dependencies manually).
> >
> >    To note, we also perform sensitivity analysis on EVs (see Appendix A), and the results show that noises or confused EVs would not significantly affect CDHRL's effectiveness.
> >
> >    > [1] Skrynnik et al., Forgetful Experience Replay in Hierarchical Reinforcement Learning from Demonstrations
> >    >
> >    > [2] Patil et al., Align-RUDDER: Learning From Few Demonstrations by Reward Redistribution
> >    >
> >    > [3] Corcoll et al., DISENTANGLING CAUSAL EFFECTS FOR HIERARCHICAL REINFORCEMENT LEARNING
> >    >
> >    > [4] Bagaria et al., OPTION DISCOVERY USING DEEP SKILL CHAINING, ICLR 2020
> >    >
> >    > [5] Li et al., Learning Subgoal Representations with Slow Dynamics, ICLR 2021
> >    >
> >    > [6] Sohn et al., Hierarchical Reinforcement Learning for Zero-shot Generalization with Subtask Dependencies. NeurIPS 2018
> >    >
> >    > [7] Chuck et al., Hypothesis-Driven Skill Discovery for Hierarchical Deep Reinforcement Learning, IROS 2020

---

### Official Review · Reviewer_E3Uf · 2022-07-12

**Rating:** 6
**Confidence:** 4
**Soundness:** 3 good
**Presentation:** 3 good
**Contribution:** 3 good

**Summary:**

The paper proposes an agent that combines causal discovery and hierarchical reinforcement learning (HRL), combined into causality-driven HRL, or CDHRL. The environment variables, which form the basis for the nodes of the structural causal model that the agent uses, are assumed to be available through an oracle. Training happens in two stages: first, the causal structure is discovered by training the agent to reach an expanding set of subgoals, after which the agent is trained on the actual task. On the two environments that are evaluated, the CDHRL method discovers reasonable causal graphs and outperforms the two baselines. Several ablations show that those results are robust to noise in the provided environment variables and limited iteration depth.

**Questions:**

- Could the authors discuss the differences between their contribution and those of Corcoll et al.? Both are aiming to introduce causality into hierarchical reinforcement learning. Not citing Corcoll et al. seems like an oversight.
- Did the authors consider environments like the ant maze in the MEGA paper? While they may not be the most interesting from a causal perspective, they would be interesting regression tests.
- Could the authors comment on the causality graphs discovered for the Eden environment? The graph for minecraft is very helpful.

**Limitations:**

- The assumption that the EVs are available through an oracle is limiting. The results presented are for cases where they are available. Discovering them in an unsupervised way can be problematic - the signal for the usefulness of candidate EVs will have to come from a combination of task reward and causality discovery success. Hence, discovering EVs is likely to require interaction with the causality discovery process, which would create a cold-start problem. The question of EV discovery is not addressed in any of the experiments. That isn’t a strict necessity for this paper, but it does limit the applicability of the method, and addressing it would make the paper stronger.
- The environment variables are assumed to be discrete. This assumption makes the exploration easier by limiting the effective number of states that need to be considered in the causal graph. Again, addressing this aspect, ideally with experiments (which would presumably take place on other environments than the ones used) would clarify the applicability domain of the proposed method.
- The causal structure of the environment is assumed to be small enough in number to constitute a nicely explorable graph - the pre-training phase runs ‘until no new causality is discovered’. This assumption won’t generally hold, and addressing the question of how much to invest in causality discovery would be valuable for understanding the generalizability of the results.


**Strengths And Weaknesses:**

- Strength: Combining causality with HRL makes lots of sense. Often, HRL approaches don’t explicitly address the causality, but assume it implicitly. Assuming the environment variables are available, providing the assumption of a causal structure as an inductive bias is as natural as transformers are for language or convnets for images.
- Strength: The interaction between the subgoal construction and the discovery process effectively induces an exploration reward, so the proposed agent not only learns about causal structure in the environment that it sees, but also ensures it sees many interesting aspects of the environment.
- Strength: the paper is well-structured and easy to read.
- Weakness: the availability of the environment variables is assumed. The authors mention a few methods aimed at discovering those, but no experiments are included where discovered environment variables are being used.
- Weakness: The results are a bit limited. Two tasks, two baselines. They make the point that in the right circumstances, this method is effective, but they don’t really address the question which circumstances are the right ones.
- Weakness: the paper is in need of a round of language editing.

---

> ### Author Response · Authors · 2022-08-02
> **(Part 1/3) Response to Reviewer E3Uf**
>
> We thank the reviewer for the insightful comments and suggestions. We would like to clarify the concerns as follows:
>
> 1. > Weakness 1: the availability of the environment variables is assumed. The authors mention a few methods aimed at discovering those, but no experiments are included where discovered environment variables are being used.
>
>    Thanks for the suggestion. There are two reasons why we assume there are available EVs:
>
>    A. First reason is that the acquisition of EV is relatively convenient in state vector-based observation, e.g. Eden and 2d-Minecraft in our experiment environments. We utilize part of the observation of the environment as EV. In Eden, we use the state values of the agent and the map as EV (including much useless noise). In 2d-Minecraft, we use the backpack observation as EV (also used by other RL methods for Minecraft [1, 2, 3]). We also conduct sensitivity analysis experiments of EV in appendix A that verify the quality of EV does not seriously affect the performance of CDHRL.
>
>    B. Another important reason is described in section 4.1 of the paper. How to discover disentangled representation for image-based observation and how to exploit the environment variables are orthogonal and both important. Our paper focuses on how to leverage causality to discover a high-quality subgoal hierarchy upon disentangled EV. So we employ an oracle way to take EV from the observation. Besides, to ensure fairness and show the effect of causality-driven discovery, we run all baselines with the  same provided EVs to compare.
>
>    > [1] Sohn et al., Hierarchical Reinforcement Learning for Zero-shot Generalization with Subtask Dependencies. NeurIPS 2018
>    >
>    > [2] Skrynnik et al., Forgetful Experience Replay in Hierarchical Reinforcement Learning from Demonstrations
>    >
>    > [3] Patil et al., Align-RUDDER: Learning From Few Demonstrations by Reward Redistribution
>
> 2. > Weakness 2: The results are a bit limited. Two tasks, two baselines. They make the point that in the right circumstances, this method is effective, but they don’t really address the question which circumstances are the right ones.
>
>    A. We attribute our method performance to the reason -- the causality can guide the agent to discover subgoals from cause to effect, in other words, from easy-to-reach to hard-to-reach. So CDHRL can automatically find high-quality subgoals and dependencies among them in hard-to-explore and sparse reward environments from scratch. Our experiments are also designed to verify such an advantage. We compare the reach frequency of subgoals in 2d-Minecraft of CDHRL with other methods (figure 3). We also compare the subgoal training efficiency of CDHRL with the advanced curriculum learning method (figure 4(c), 4(d)).
>
>    B.We add an extra advanced baseline called LESSON [1], which selects slowly changed features in the state as subgoals. Its performance is also less than our method because such a strategy needs trajectories that reach subgoals; however, it cannot meet in hard-to-explore environments (LESSON [1] is verified in the easy-to-explore ant-maze environment).
>
>    (see https://ibb.co/YtgPxts (Eden) and https://ibb.co/CHMLVck (2d-Minecraft))
>
>    > [1] Sohn et al., Hierarchical Reinforcement Learning for Zero-shot Generalization with Subtask Dependencies. NeurIPS 2018
>
> 3. > Weakness 3: the paper is in need of a round of language editing.
>
>    Thanks for your suggestions, we have revised our writings.
>
> 4. > Question 1: Could the authors discuss the differences between their contribution and those of Corcoll et al.? Both are aiming to introduce causality into hierarchical reinforcement learning. Not citing Corcoll et al. seems like an oversight.
>
>    Thanks for the referring. We include discussion in Section Related works and added this work as a reference.
>
>    Corcoll et al. [1] introduce the causality into hierarchical reinforcement learning in a unidirectional way. Our work not only introduces the
>    causality to guide subgoal discovery, but also leverages the sub-goal policy to enhance the causality discovery in environments, which
>    forms the boosting loop between causality and HRL. The loop is the key insight and contribution of CDHRL to address the issue of low
>    exploration efficiency for hard-to-reach subgoals in complex HRL environments. Without a loop, Corcoll et al. only consider simple one-
>    step causality (action to state) and cannot handle the tasks in complex environments with the deep subgoal hierarchy (They verify their
>    method in an environment that only contains a few variables and one-step causality).
>
>    > [1] Corcoll et al., DISENTANGLING CAUSAL EFFECTS FOR HIERARCHICAL REINFORCEMENT LEARNING

---

> > ### Author Response · Authors · 2022-08-02
> > **(Part 2/3) Response  to Reviewer E3Uf**
> >
> > 5. > Question 2:  Did the authors consider environments like the ant maze in the MEGA paper? While they may not be the most interesting from a causal perspective, they would be interesting regression tests.
> >
> >    We do not consider environments like Robot control and Maze. As described in the Related works (lines 86-88), because these environments contain simple relations and easy-to-reach subgoals, the random-driven paradigm of the existing HRL method is sufficiently good to discover high-quality subgoals. Our method aims to discover high-quality subgoals in complex and hard-to-explore environments.
> >
> > 6. > Question 3: Could the authors comment on the causality graphs discovered for the Eden environment? The graph for minecraft is very helpful.
> >
> >    Limited by the page constraints, we show the causality graph discovered in Eden in Appendix. Compared with Minecraft, the causality graph discovered in Eden is more accurate since the causality in Eden is more sparse and straightforward.
> >
> >    (see https://ibb.co/1Lc7dYh)
> >
> > 6. > Limitation 1:  The assumption that the EVs are available through an oracle is limiting. The results presented are for cases where they are available. Discovering them in an unsupervised way can be problematic - the signal for the usefulness of candidate EVs will have to come from a combination of task reward and causality discovery success. Hence, discovering EVs is likely to require interaction with the causality discovery process, which would create a cold-start problem. The question of EV discovery is not addressed in any of the experiments. That isn’t a strict necessity for this paper, but it does limit the applicability of the method, and addressing it would make the paper stronger.
> >
> >    In fact, the Environment Variables (EV) in our paper are restricted as disentangle factors in observation other than extra information. Also,  EV are not the perfect abstract representation states that experts provide.
> >
> >    **For environments providing state vector-based observation** (which cover a broad category of environments, such as Atari [5], Mujoco [6], 2d-Minecraft [1], ...), obtaining "Environment Variables" (EV) is relatively convenient. The state vector-based observation can be directly transformed into the EV vector since information of different dimensions is disentangled (In our experiment, we take all disentangled factors in observation as the EV, including many noise and useless variables). Many HRL methods [1, 4, 7, 8, 9] commonly use them as the standard inputs. For example, LESSON[4] assumes that each input state dimension represents an independent feature. DSC[8] uses disentangled factors like position, orientation, linear velocity, rotational velocity, and a has_key indicator as the state space.
> >
> >    **For environments with image-based observation,** finding EV has been largely studied, like CausalVAE [10] and DEAR [11]. Considering that our paper focuses on how to build the subgoal hierarchy instead of finding EV, we do not include that part in this paper. Thanks for the great advice, and we will integrate the EV exploring in our CDHRL paradigm for environments with image-based observations in the future.
> >
> >    To note, we also perform sensitivity analysis on EVs (see Appendix A), and the results show that noises or confused EVs would not significantly affect CDHRL's effectiveness.
> >
> >    > [1] Sohn et al., Hierarchical Reinforcement Learning for Zero-shot Generalization with Subtask Dependencies. NeurIPS 2018
> >    >
> >    > [2] Skrynnik et al., Forgetful Experience Replay in Hierarchical Reinforcement Learning from Demonstrations
> >    >
> >    > [3] Patil et al., Align-RUDDER: Learning From Few Demonstrations by Reward Redistribution
> >    >
> >    > [4] Li et al., Learning Subgoal Representations with Slow Dynamics, ICLR 2021
> >    >
> >    > [5] Anand et al., Unsupervised State Representation Learning in Atari.  NeurIPS 2019
> >    >
> >    > [6] Todorov et al., MuJoCo: A physics engine for model-based control.
> >    >
> >    > [7] Corcoll et al., DISENTANGLING CAUSAL EFFECTS FOR HIERARCHICAL REINFORCEMENT LEARNING
> >    >
> >    > [8] Bagaria et al., OPTION DISCOVERY USING DEEP SKILL CHAINING, ICLR 2020
> >    >
> >    > [9] Chuck et al., Hypothesis-Driven Skill Discovery for Hierarchical Deep Reinforcement Learning, IROS 2020
> >    >
> >    > [10] Yang et al., Causalvae: Disentangled representation learning via neural structural causal models. CVPR, 2021
> >    >
> >    > [11] Shen et al., Disentangled generative causal representation learning

---

> > > ### Author Response · Authors · 2022-08-02
> > > **(Part 3/3) Response to Reviewer E3Uf**
> > >
> > > 8. > Limiattion 2:  The environment variables are assumed to be discrete. This assumption makes the exploration easier by limiting the effective number of states that need to be considered in the causal graph. Again, addressing this aspect, ideally with experiments (which would presumably take place on other environments than the ones used) would clarify the applicability domain of the proposed method.
> > >
> > >    The limitation of discrete EV is mainly because of the form of variable change functions. In other words, we need to define more complex function forms for continuous variables. We leave broadening the application scope of CDHRL as our future work. Consider the acquisition of EV, because obtaining EV is relatively convenient in environments with state vector-based observation, which makes our method applicable in broad scenarios. Only in environments with purely image-based observation, the disentangled representation encoder is needed, but the methodology of subgoal discovery is generic. We have added this limitation to the Conclusion of the paper.
> > >
> > > 9. > Limitation 3:  The causal structure of the environment is assumed to be small enough in number to constitute a nicely explorable graph - the pre-training phase runs ‘until no new causality is discovered’. This assumption won’t generally hold, and addressing the question of how much to invest in causality discovery would be valuable for understanding the generalizability of the results.
> > >
> > >    As mentioned above, the environment variables are restricted as disentangle factors in observation. Moreover, the causality is sparse (a generic assumption in causal discovery) and non-cyclic. The convergence time is only determined by the depth of the causality graph, which would not be too deep in most environments. Thanks for your suggestion, and we will consider doing reasonable constraints with the depth or discovery iteration rounds of the causality graph in the future.

---

### Author Response · Authors · 2022-08-07
**Follow-up**

Dear Reviewers,

We sincerely thank the reviewers for their time and diligence in reviewing our submission. We have included detailed responses for each reviewer and look forward to following up to see if our response has addressed your concerns or if you have any further questions.

Thank you again for the reviews!

---

### Author Response · Authors · 2022-08-09
**Letter to ACs and all reviewers**

Dear ACs and all reviewers,

We sincerely thank the reviewers for their time and diligence in reviewing our submission. Since the deadline of discussion is approaching, we look forward to follow-up discussions needed from reviewers.

So far, we received three positive scores (6s) and one negative score (4). We hope our additional experiments and detailed response have convinced the reviewer on the merits of our submission.

Reviewer 6RRy (giving a score of 4) doubts the novelty and necessity of introducing causality (rather than other relationships) into Hierarchical RL, and concerns with the difference with other graph-based methods (for example, graph-based imitation learning methods). Some misunderstandings about our algorithms might exist. We restate the novelty and irreplaceability of leveraging causality to achieve "smarter to explore in a smaller space" (as said in 6RRy's review). So that our method can discover high-quality subgoals from easy to hard in hard-to-explore environments. Besides, we also explain the difference between our method and other graph-based imitation learning methods [2, 3] that rely on successful demonstrations and in a hard-encoded way to construct subtask chain. Leveraging the graph structure for finding skills is the common design philosophy for HRL techniques. However, our method automatically finds high-quality subgoals and dependencies of subgoals from scratch.

Reviewer K7mE (giving a score of 6) is concerned about the baseline methods we compared. We add a new comparison with LESSON[4] in the revised paper. Results show that LESSON performs similarly compared with old baseline HAC, while CDHRL outperforms them.

Reviewer E3Uf, 6RRy, and K7mE are concerned that obtaining Environment Variables (EV) may limit the application of our method. We clarify the obtaining of EV and emphasize that the application of our method is broad.

> For environments providing state vector-based observation (which cover a broad category of environments, such as Atari [3], Mujoco [4], 2d-Minecraft [5], ...), obtaining "Environment Variables" (EV) is relatively convenient. For CDHRL, EV are disentangled factors in the environment observation rather than the perfect abstract representation states that experts provide. The state vector-based observation can be directly transformed into the EV vector since information of different dimensions is disentangled (In our experiment, we take disentangled factors in observation as the EV, including many noise and useless variables). Many HRL methods [2, 5, 6, 7, 8] commonly use them as the standard inputs. For example, LESSON [2] assumes that each input state dimension represents an independent feature. DSC [7] uses disentangled factors like position, orientation, linear velocity, rotational velocity, and a has_key indicator as the state space. For environments with image-based observation, finding EV has been largely studied, like CausalVAE [9] and DEAR [10]. Considering that our paper focuses on how to build the subgoal hierarchy instead of finding EV, we do not include that part in this paper. Thanks for the great advice, and we will integrate the EV exploring in our CDHRL paradigm for environments with image-based observations in the future.
>
> To note, we also perform sensitivity analysis on EVs (see Appendix A), and the results show that noises or confused EVs would not significantly affect CDHRL's effectiveness.
>
> [1] Sohn et al., Hierarchical Reinforcement Learning for Zero-shot Generalization with Subtask Dependencies. NeurIPS 2018
>
> [2] Skrynnik et al., Forgetful Experience Replay in Hierarchical Reinforcement Learning from Demonstrations
>
> [3] Patil et al., Align-RUDDER: Learning From Few Demonstrations by Reward Redistribution
>
> [4] Li et al., Learning Subgoal Representations with Slow Dynamics, ICLR 2021
>
> [5] Anand et al., Unsupervised State Representation Learning in Atari. NeurIPS 2019
>
> [6] Todorov et al., MuJoCo: A physics engine for model-based control.
>
> [7] Corcoll et al., DISENTANGLING CAUSAL EFFECTS FOR HIERARCHICAL REINFORCEMENT LEARNING
>
> [8] Bagaria et al., OPTION DISCOVERY USING DEEP SKILL CHAINING, ICLR 2020
>
> [9] Chuck et al., Hypothesis-Driven Skill Discovery for Hierarchical Deep Reinforcement Learning, IROS 2020
>
> [10] Yang et al., Causalvae: Disentangled representation learning via neural structural causal models. CVPR, 2021
>
> [11] Shen et al., Disentangled generative causal representation learning

In the response, we have clarified these misunderstandings and responded to all the questions with qualitative and quantitative explanations.

We hope that all reviewers could take a close look at our response and find it convincing.

Thank you for your time and help!



Best,

Authors of submission “Causality-driven Hierarchical Structure Discovery for Reinforcement Learning”

---

### Meta-Review · Area_Chair_3Ejk · 2022-08-25

**Recommendation:** Accept
**Confidence:** Certain

**Metareview:**

After a strong rebuttal from the authors and an extensive discussion among the reviewers, I believe the paper's pros outweigh its cons and this paper will be a valuable contribution to NeurIPS. I recommend it for acceptance and encourage the authors to address the reviewers comments for the camera-ready version of the paper - specifically please add the posted clarifications and experiments from the rebuttal and justify the limitation of available environment variables.


**Award:**

No

---

### Decision · Program_Chairs · 2022-09-14

Accept